# Why maternal continuum of care remains low in Northwest Ethiopia? A multilevel logistic regression analysis

**Tesfahun Hailemariam**[1,2]\*, **Asmamaw Atnafu**[3], **Lemma Derseh Gezie**[4], **Binyam Tilahun**[1]

**1** Department of Health Informatics, Institute of Public Health, College of Medicine and Health Sciences, University of Gondar, Gondar, Ethiopia, **2** Department of Health Informatics, College of Health Sciences, Hawassa, Ethiopia, **3** Department of Health System and Policy, Institute of Public Health, College of Medicine and Health Sciences, University of Gondar, Gondar, Ethiopia, **4** Department of Epidemiology and Biostatistics, Institute of Public Health, College of Medicine and Health Sciences, University of Gondar, Gondar, Ethiopia

\* tesfahunhailemariam@gmail.com

**Data Availability Statement:** All relevant data are within the paper and its Supporting Information files.

## Abstract

### Background

Non-adherence to the maternal continuum of care remains a significant challenge. Though early initiation and continuum of care are recommended for mothers' and newborns' well-being, there is a paucity of evidence that clarify this condition in resource-limited settings. This study aimed to assess the level of women's completion of the maternal continuum of care and factors affecting it in Northwest Ethiopia.

### Methods

A community-based cross-sectional study was conducted from October to November, 2020. Data were collected from 811 women who had a recent history of birth within the past one year. The random and fixed effects were reported using an adjusted odds ratio with a 95% confidence interval. The p-value of 0.05 was used to declare significantly associated factors with women's completion of the maternal continuum of care.

### Results

The study revealed that 6.9% (95%CI: 5.3–8.9%) of women were retained fully on the continuum of maternal care, while 7.89% of women did not receive any care from the existing health-care system. Attending secondary and above education (AOR = 3.15; 95%CI: 1.25,7.89), membership in the women's development army (AOR = 2.91; 95%CI: 1.56,5.44); being insured (AOR = 2.59; 95%CI: 1.33,5.01); getting health education (AOR = 2.44; 95%CI: 1.33,4.45); short distance to health facility (AOR = 4.81; 95%CI: 1.55,14.95); and mass-media exposure (AOR = 2.39; 95%CI: 1.11,5.15) were significantly associated with maternal continuum of care.

### Conclusions

The maternal continuum of care is low in rural northwest Ethiopia compared to findings from most resource-limited settings. Therefore, the existing health system should consider

**Funding:** Doris Duke Charitable Foundation under grant number 2017187.

**Competing interests:** The authors declare that they have no competing interests.

multilevel intervention strategies that focus on providing maternal health education, facilitating insurance mechanisms, encouraging women's participation in health clubs, and ensuring physical accessibility to healthcare facilities to be more effective in improving maternal health services.

## Introduction

Every day, about 810 women die due to pregnancy and childbirth-related complications across the world [1]. Annually, 295,000 women die as a result of pregnancy and childbirth-related complications, of which 94% occur in low- and middle-income countries. Statistics, according to WHO 2019 report on trends in maternal mortality estimates from 2000–2017, show that the probability that a 15-year-old woman will die eventually from a maternal cause is estimated as 1 in every 90 women globally, while 1 in every 5400 and 45 in developed and developing countries, respectively [1]. Likewise, the life time risk of maternal death is recorded as 1 in every 37 in Sub-Saharan Africa [1]. Almost all pregnancy-related deaths are associated with complications during pregnancy and childbirth, and nearly 75% of these deaths could be avoided if women were kept on the maternal continuum of care [2]. According to the 2019 report, from global figures, approximately 14,000 maternal deaths and 1/55 lifetime risk of maternal mortality were reported in Ethiopia [1]. Despite recent progress, Ethiopia has continued to suffer from an unacceptable number of maternal deaths [3].

Improving maternal health has been a global public health priority [4, 5]. The maternal continuum of care through pregnancy to postpartum is a rallying call to make a woman's life safe and joyful [6]. It is a proven intervention for maternal mortality reduction and is efficient for making a woman more productive in her life [7], as it links a woman to the maternal healthcare services provision centers such as home, community, and health facilities [6]. The World Health Organization (WHO) suggested at least four Antenatal Care (ANC) visits and one or more postnatal visits within two days of delivery to improve maternal health and birth outcome [8, 9].

ANC is a vital component of the maternal continuum of care for women and is believed to be the basis for women and newborns to survive and thrive [10, 11]. Early initiation and proper ANC follow-up reduce adverse pregnancy outcomes in both mother and baby since it promotes early identification of pregnancy-related complications and provides an opportunity for a woman to discuss with a healthcare provider about pregnancy and health behavior [12]. Healthcare facility delivery with the assistance of a Skilled Birth Attendant (SBA) is another strategy for reducing maternal morbidity and mortality [9], as births carried out in homes under unhygienic conditions could lead to a high incidence of maternal death [13]. Receiving Postnatal Care (PNC) from a healthcare provider at the recommended time with its appropriate content also prevents complications that could arise after childbirth and helps a mother to get comprehensive health education [9].

With regard to the health sector's mission in Ethiopia, a three-tiered healthcare delivery system has been put in place to facilitate the realization of the health system's goals; at the primary level of the healthcare delivery system, Health Extension Workers (HEWs) are designated for consultation and service provision in the areas of family planning, ANC, PNC, and child health services [14]. Moreover, the government's initiatives and commitment towards the maternal continuum of care indicate there are identified intervention areas to strengthen maternal health programs [15]. Despite the improvement in maternal healthcare utilization, however,

completion of the maternal continuum of care is very low [16] and improving maternal health remains a major challenge for the health system in Ethiopia [3].

According to past studies, nearly four out of every ten women did not receive ANC; seventeen percent of women in Ethiopia had their first prenatal care visit before the fourth month of pregnancy [17] with 56% of women giving birth in a health facility after receiving four or more ANC visits, and 17% received postnatal visits within 48 hours after birth [9].

According to the body of literature, both individual and community-level factors are influencing maternal healthcare utilization [18–20]. Though there was promising progress in maternal healthcare utilization, and it is a prioritized agenda in Ethiopia [21], the available evidence on women's completion of the maternal continuum of care is inconclusive. Understanding the existing point of women's completion of the maternal continuum of care and identifying contextual factors that limit women's completion of the maternity care continuum helps to be effective in the implementation of maternal health programs. Considering the hierarchical structure of women with a child less than one year (level 1) nested within a health center (level 2), it is appropriate to assess the point of women's retention on the maternal continuum of care. Therefore, this study aimed to determine the level of women's completion of maternal continuum of care and identify individual and community-level factors affecting it in northwest Ethiopia.

## Methods

### Study design and setting

A community-based cross-sectional study was conducted in 11 clusters in Wogera and Gondar Zuriya districts in the central Gondar zone of Amhara National Regional State from October to November 2020. The districts are located 658km from Addis Ababa, the capital city of Ethiopia. There were 16 health centers and 88 health posts in the districts. The total population was 524, 907 (female = 260879 and male = 264028) at the time of the survey, of which 122,303 women in the reproductive age group (15–49) and 16745 were surviving infants (Central Gondar Zone Health Bureau report-unpublished).

### Study population

All women in the study districts who gave birth within the past one year prior to data collection were the source population, and randomly selected women who were permanent residents of the selected kebeles and willing to participate in the study were the study population.

### Sample size determination

For the single population proportion formula, the assumptions used were proportion of focused ANC p = 39.9%, proportion of SBA after completing at least 4 ANC visits p = 31.1%, and proportion of women retained in the continuum of maternal care p = 12.1% [22]. This formula considers 95% CI, margin of error = 5%, design effect of 2 and 10% non-response rate, the sample sizes were 811,724, and 360, respectively. Double population proportion formulas were considered in estimating different sample sizes, considering factors affecting maternal health service utilization. The sample size was computed using the STAT-CALC program of Epi-info version 7.0. software using the following assumptions: 5% level of significance (two-sided), 80% power, a 1:1 ratio of exposed to non-exposed maternal health service utilization, 49.7% of the outcome in the exposed group and 50.3% of the outcome in the unexposed group, and considering being a model household in the community as an exposure variable for good maternal health service utilization [15], a design effect of 2, and a 10% non-response

rate, n = 559. Of the different sample sizes estimated, the largest sample size, 811, was obtained from a single population formula and considered in the study.

## Sampling procedure

A two-stage sampling technique was employed to reach study participants. Since the districts have fifteen health centers, eleven health centers (clusters), kebeles at each cluster were selected randomly with systematic sampling. Households were selected from a sampling frame developed using a family folder of HEWs, which contained a list of all births for rural women. The sampling interval was obtained by dividing the source population by the total sample size estimated. The first woman was chosen from a sampling frame in the family folder using a simple random sampling technique, and the random selection was carried out using the random between functionality in Microsoft Excel. Using this technique, the required sample size was obtained, and a total of 811 eligible women were interviewed for the study and included in the analysis.

## Study variables and measurement

**Response variable.**   Women's completion of the maternal continuum of care is categorized as "yes" if the women received at least 4 ANC visits, SBA, and PNC visits within 48 hours after birth, and "no" if otherwise.

**Explanatory variables.**   Ranges of predictor variables were selected and grouped as individual and community level variables based on previous literature.

**Individual level factors.**   Individual level variables were age of the respondents, marital status of the respondents, occupation, education status of respondents, education status of their partners, pregnancy intention, distance from home to facility, household wealth index, being a member of community-based health insurance, mass-media exposure, HEWs' home visits during pregnancy, being a member of Women's Development Army (WDA), and getting health education from a nearby health post during pregnancy.

**Community level factors.**   Community and cluster level variables were aggregated from individual and cluster level data such as health facility readiness, the average distance from their village to a health facility, wealth status in the cluster, and mass-media exposure. Health facility readiness for service provision was assessed using the WHO service availability and readiness assessment guideline [23]. Aggregated clusters were categorized as low if the proportion of clusters' readiness for service provision was 0–72% whereas high if the proportion was 73–100%. The household wealth index median value was three, and the community value was classified as low if the median value of the community is below three and high if otherwise (middle and above). The mass-media exposure status of the community was categorized as low if the proportion of exposure status was 0–17%, and whereas high if the proportion of mass media exposure status in the community was 18–100%. Besides, the walking distance from home to health facility was near if the aggregated proportion was 0–71%, and far if the aggregated proportion of walking distance was 72–100%.

## Data collection instrument and procedures

The instrument was developed in English and translated into the local language (Amharic). Since experts' views were sought for the psychometric properties of the face and content validity, experts with health management system expertise, health informatics professionals, midwifery professionals, gynecologist and obstetrician, and emergency officers at obstetrics and gynecology were invited to review the relevance of each question in the instrument. The tool was revised according to the experts' views and was piloted out of the selected clusters and

validated before data collection. The proportion of the rated item and scale content validity index was computed, and the experts' rated item content validity index and scale level validity index were 0.98 and 0.81, respectively. Data collectors and supervisors were trained on the instrument. The household data were collected using an interviewer-administered questionnaire via face-to-face interviews.

## Operational definition

**Maternal continuum of care.** It is the continuity of care throughout pregnancy (utilization of at least 4 ANC visits, SBA, and PNC within 48 hours after birth) [6].

## Data management and statistical analysis

The data were entered into Epidata and analyzed with Sata14. For wealth index assessment, principal component analysis was employed to reduce data that measure the same construct together, and the data were recoded into binary variables. Variables with low frequencies were combined, and frequencies greater than 95% and less than 5% were excluded from the analysis. Under the component loading, there were twenty-one variables correlated with measuring the wealth index, such as having one's own house, toilet, electricity, kitchen, charcoal, television, radio, modern bed, cotton, mobile phone, farming land; having animals in the house, such as cows, horse, donkey, goat, sheep, chickens; having a separate room for animals; having a beehive, and having a bank account number with deposited Birr (Ethiopian currency). The wealth quantile was categorized into five categories, such as poorest, poor, middle, richer, and richest.

Bivariable multilevel logistic regression was used for each factor against the maternal continuum of care without controlling the effect of other explanatory variables. For multivariable multilevel logistic regression analysis, factors at both individual and cluster levels with a p-value <0.25 were considered candidate variables. The fixed-effect of individual and cluster level variables was reported using the Crude Odds Ratio (COR) with a 95% confidence interval.

During model building, four models were built to estimate both the fixed effects of the individual and community-level factors and the random effects between-community variation on the women's completion of the maternal continuum of care. Model I was built without any explanatory predictors to examine the random effect of cluster variation by using Intra-Class Correlation (ICC) to justify the application of multilevel analysis in this study. Therefore, random parameters in this model were used as a benchmark to compare parameters of successive models (Model II, Model III, and Model IV) by looking at the decline of the ICC value [24]. Model II was built to examine the contributions of individual-level factors. Model III was fitted to determine between-cluster variations. Finally, Model IV (individual and cluster-level factors model) was built by combining both individual and cluster-level factors simultaneously by controlling for the effect of other predictors. The measure of association was reported as Adjusted Odds Ratio (AOR) with a 95% CI. The p-value < 0.05 was used to identify factors significantly associated with women's completion of the maternal continuum of care. The random effect was presented using ICC; thus, in all models, ICC and its change from the null model were examined. The Proportional Change in Variance (PCV) was computed with reference to the null model to examine the relative contribution of level II, III, and IV predictors in explaining the odds of women completing the maternal continuum of care.

To estimate the goodness-of-fit of the adjusted final model, the Akaike information criteria (AIC) and loglikelihood model were used in comparison with other models. The multicollinearity effect was checked by using the mean of the variation inflation factor (VIF) value at a cut-off point of 10, and it indicated that there was no multicollinearity effect among predictor variables [25]. The interaction effect of the variables was checked by creating a new variable,

and the created new variable, or product term, became either statistically significant or not at p-value<0.05.

## Ethical considerations

Verbal consent was obtained and participants were informed about the objective, importance of the study, procedure and duration, risk and discomfort, benefits of participating in the study, confidentiality, and the right to refuse or withdraw during data collection. Study approval and ethical clearance were obtained from the University of Gondar ethical review board (R.NO. V/P/RCS/05/2020). A formal letter of approval was taken from Amhara national regional health state bureau and central Gondar zonal health department. For participants age <18 verbal informed consent was taken from their parents and assent obtained from the minor/participant. And it was approved by the ethical review committee of the institute of public health on behalf of IRB of University of Gondar. After obtaining the relevant information, participants were counselled on the benefits of attending maternal health care services and the consequences of missing maternal health care services. The COVID-19 protocol was maintained throughout the study.

## Results

### Sociodemographic and reproductive characteristics of the study participants

A total of 811 women participated in the study, with a response rate of 100%. The minimum and maximum ages of the respondents were 15 years and 48 years, with a mean age of 28 years. The majority of the participants were married, 790(97.4%), and housewives, 801(98.8%). About two-thirds, 507(61.7%), of participants did not attend education. The proportion of women who intended the current pregnancy was 642 (79.2%). The proportion of women was nearly equal across the wealth quantiles at household level, with 163(20.1%) the poorest, 162 (20.0%) poorer, 163 (20.1%) middle, 161(19.8%) richer, and 162(20.0%) the richest Table 1.

Being a member of community-based health insurance and being a member of a WDA in the kebeles was 394 (48.6%) and 260 (32.1%), respectively. Study participants who were visited by HEWs during the current pregnancy were 361 (44.5%) and those who got health education from a nearby health post were 314 (38.7%) Table 1.

### Community level characteristics of the study participants

Women's retention in the maternal continuum of care was significantly associated with clusters (health centers) (chi-square = 28.19, p-value = 0.001). In this study, the wealth index at the community level indicated that 611 (75.3%) of the participants had high wealth status. The level of health facility readiness for ANC service provision was 619 (76.3%). The proportion of study participants who had high media exposure at the community level was 372 (45.9%). Among the participants, most respondents, 615(75.8%), had a walking distance of less than or equal to five kilometers from their home to a health facility. Bivariable multilevel logistic regression analysis indicated that media exposure in the community and distance from home to health facility were significantly associated factors with the maternal continuum of care Table 2.

### Maternal continuum of care

In this study, women who attended at least 4 ANC visits, SBA, and PNC visits within 48 hours after birth were considered during the analysis. Any ANC visit: Women who received at least one antenatal care visit from the existing healthcare system. At least 4 ANC visits: Women who received at least 4 ANC visits. Retention on SBA: This is the continuity of care from at

**Table 1. Sociodemographic and reproductive characteristics of study participants in Northwest Ethiopia, 2020 (n = 811).**

| Variables | Category | Maternal continuum of care | | (p-value) |
|---|---|---|---|---|
| | | Yes(%) | No(%) | |
| **Age group of respondents (years)** | 15–24 | 16(7.0) | 211(93.0) | 0.935 |
| | 25–34 | 28(7.1) | 366 (92.9) | |
| | 35 and above | 12(6.3) | 178(93.7) | |
| **Marital status** | Married | 54(6.8) | 736(93.2) | 0.632 |
| | Single/divorced/widowed /separated | 2(9.5) | 19(90.5) | |
| **Occupation** | Housewife | 55(6.9) | 746(93.1) | 0.698 |
| | Employee/laborer/merchant | 1(10.0) | 9(90.0) | |
| **Education level of respondents** | Did not attend education | 23(4.6) | 477(95.4) | <0.001 |
| | Primary education | 17(7.9) | 198(92.1) | |
| | Secondary and above | 16(16.7) | 80(83.3) | |
| **Education level of husbands** | Did not attend education | 27(5.2) | 496(94.8) | <0.001 |
| | Primary education | 15(7.3) | 190(92.7) | |
| | Secondary and above | 14(16.9) | 69(83.1) | |
| **Pregnancy intention** | Yes | 49(7.6) | 593(92.4) | 0.111 |
| | No | 7(4.1) | 162(95.9) | |
| **Distance home to health facility** | >5km | 23(9.8) | 212(90.2) | 0.039 |
| | < = 5km | 33(5.7) | 543(94.3) | |
| **Household wealth index** | Poorest | 11(6.7) | 152(93.3) | 0.446 |
| | Poorer | 6(3.7) | 156(96.3) | |
| | Middle | 13(8.0) | 150(92.0) | |
| | Richer | 12(7.5) | 149(92.5) | |
| | Richest | 14(8.6) | 148(91.4) | |
| **Being insured** | Yes | 41(10.3) | 353(89.6) | < .000 |
| | No | 15(3.6) | 402(96.4) | |
| **Membership in WDA** | Yes | 35(13.5) | 225(86.5) | < .000 |
| | No | 21(3.8) | 530(96.2) | |
| **HEWs home visiting** | Yes | 31(8.6) | 330(91.4) | 0.091 |
| | No | 25(5.6) | 425(94.4) | |
| **Mass media exposure** | Yes | 15(10.7) | 125(89.3) | 0.051 |
| | No | 41(6.1) | 630(93.9) | |
| **Getting health education at health post level** | Yes | 29(9.2) | 285(90.8) | 0.037 |
| | No | 27(5.4) | 470(94.6) | |

least 4 ANC visits to skilled delivery. Retention on PNC: This is about the continuity of care from ANC to SBA and PNC within the first 48 hours after delivery. The outcome variable was "yes" if a woman received ANC to SBA and PNC within the first 48 hours after birth and "no" if otherwise. Accordingly, the rates of any ANC visit during pregnancy, at least 4 ANC visits, and SBA after ANC4+ completion were 86.8% (95% CI: 84.3–89.1), 39.6% (95% CI: 36.2–43), and 29.1% (95% CI: 26–32.4), respectively. The proportion of women who were retained fully on the maternal continuum of care during pregnancy was 6.9% (95%CI: 5.3–8.9) Fig 1.

## Multivariable multilevel logistic regression analysis of maternal continuum of care

During analysis, four mixed-effect regression models were built to predict the maternal continuum of care based on the different exposure variables at individual and community levels

**Table 2. Community level characteristics of respondents in Northwest Ethiopia, 2020 (n = 811).**

| Variables | Category | Maternal continuum of care | | (p-value) |
|---|---|---|---|---|
| | | Yes(%) | No(%) | |
| **Health facilities** | Ambageorgis | 16(15.2) | 89(84.8) | 0.001 |
| | Gedebiye | 11(11.6) | 84(88.4) | |
| | Birra | 3(5.0) | 57(95.0) | |
| | Tirgosgie | 6(9.0) | 61(91.0) | |
| | Woybey | 5(10.9) | 41(89.1) | |
| | Dergaj | 3(7.1) | 39(92.9) | |
| | Miniziro | 0(0.0) | 104(100.0) | |
| | Maksegnit | 3(3.5) | 82(96.5) | |
| | Lamba | 2(2.5) | 78(97.5) | |
| | Enfranz | 4(4.3) | 88(95.7) | |
| | Abawarka | 3(8.6) | 32(91.4) | |
| **Wealth Index** | High | 45(8.2) | 506(91.8) | 0.039 |
| | Low | 11(4.2) | 249(95.8) | |
| **Health facility readiness** | High | 48(7.8) | 571(92.2) | 0.087 |
| | Low | 8(4.2) | 184(95.8) | |
| **Mass-media exposure** | High | 34(9.1) | 338(90.9) | 0.021 |
| | Low | 22(5.0) | 417(95.0) | |
| **Walking distance from home to health facility** | >5km | 4(2.0) | 192(98.0) | 0.002 |
| | < = 5km | 52(8.5) | 563(91.5) | |

Table 3. The intercept-only model was run without any predictor to test the random effect between cluster variation on the maternal continuum of care. An estimate of ICC was 12% (95% CI: 3%, 35%), implying that 12% of the variation in the maternal continuum of care was due to cluster-level factors and 43.5% was due to differences across clusters, and this variation was significant ($\tau$ = 0.43, p 0.000).

At the individual level (Model II), predictors such as women with secondary and above education status, being a member of the WDA, being a member of community-based health insurance, getting health education at the health post level, distance from home to health facility, and husband's secondary and above education level were statistically associated with women's completion of the maternal continuum of care. ICC in Model II indicated that 10% of the difference in women's completion of the maternal continuum of care was attributed to cluster variability.

The finding of this study in Model III showed that exposure to mass media and walking distance of less than or equal to five kilometers from home to a health facility were statistically significant in women's completion of the maternal continuum of care.

Finally, after controlling all factors, the full model (Model IV) was developed, including individual and cluster-level factors simultaneously. The findings indicated that there was a substantial reduction in variance in predicting women's completion of the maternal continuum of care. About 93% of women's completion of the maternal continuum of care in the clusters was explained in the last model where the lowest Akaike information criteria and loglikelihood were observed (362.62 and 168.31, respectively). The odds of adhering to the maternal continuum of care were 3.15 times (AOR = 3.15; 95%CI: 1.25, 7.89) higher among women with secondary and above education as compared to their counterparts with lower-education. Women who were members of a WDA in the kebeles were 2.91 times (AOR = 2.91; 95%CI: 1.56, 5.44) more likely to complete the maternal continuum of care as compared to

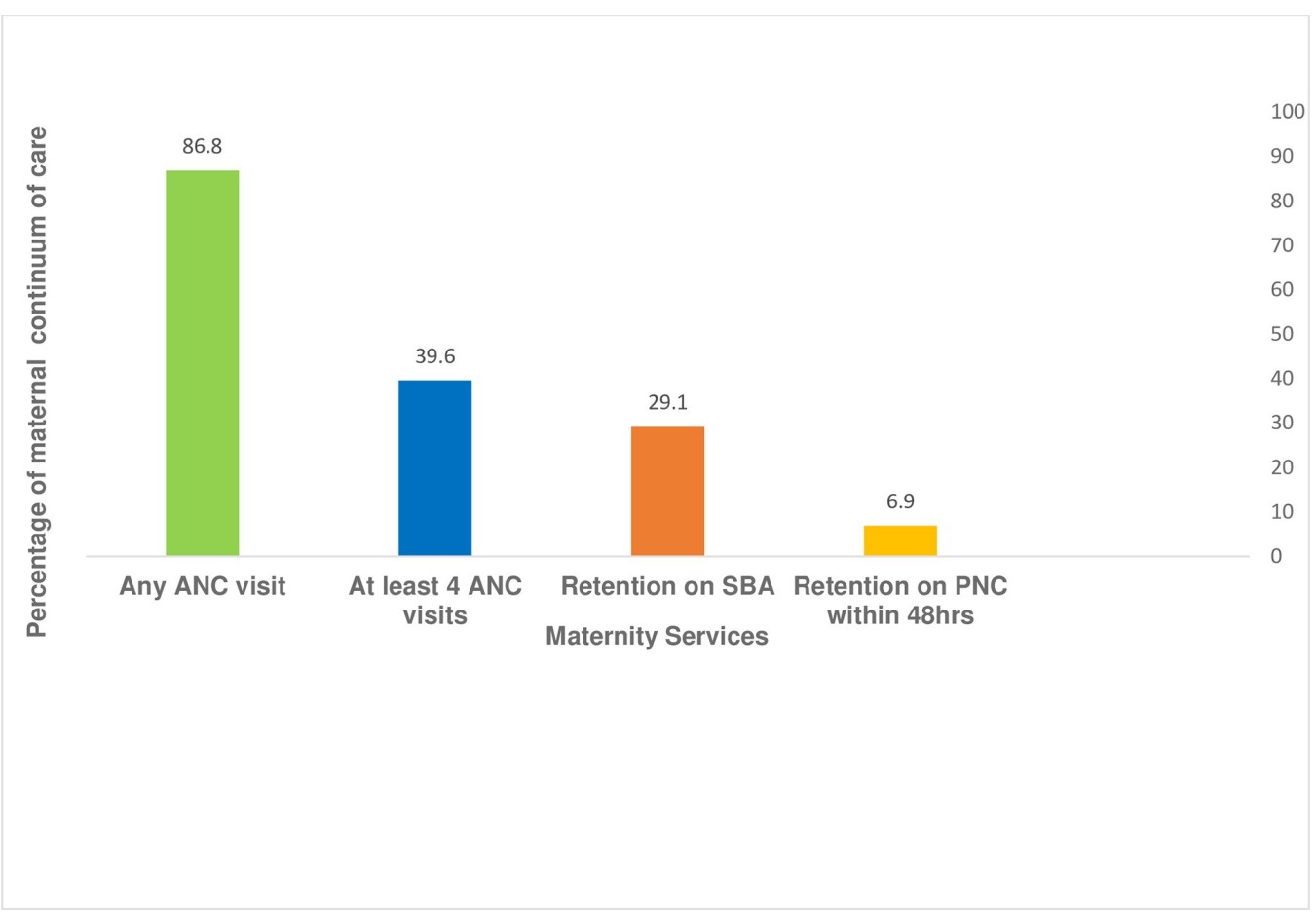

**Fig 1. Maternal health service utilization in completion of maternal continuum of care from ANC to PNC care within 48hrs after birth in Northwest Ethiopia, 2020.**

women who were not members of a WDA. Women insured in community-based health schemes were 2.59 times more likely to complete the maternal continuum of care than their counterparts (AOR = 2.59; 95%CI: 1.33, 5.01).

Our study found that women who got health education from a health post during the pregnancy period were 2.44 times (AOR = 2.44; 95%CI: 1.33, 4.45) more likely to complete the maternal continuum of care as compared to women who did not get health education from a nearby health post. Furthermore, women who walked less than or equal to five kilometers from home to health facility were 4.81 times (AOR = 4.81; 95%CI: 1.55,14.95) more likely to complete the maternal continuum of care than those who walked more than five kilometers. Finally, women who had been exposed to mass media were 2.39 times (AOR = 2.39; 95%CI: 1.11, 5.15) more likely to complete the maternal continuum of care as compared to their counterparts Table 3.

## Discussion

We conducted a study on women's completion of the maternal continuum of care among women in rural areas of Northwest Ethiopia. Our study showed that nearly one out of every fourteen women completes a maternal continuum of care, while one out of every thirteen women does not receive maternal healthcare from the existing health system. The final model

**Table 3. Multivariable multilevel logistic regression analysis of predictors of women's completion of maternal continuum of care in Northwest Ethiopia, 2020 (n = 811).**

| Variables | Category | COR (95%CI) | Model I | Model II | Model III | Model IV |
|---|---|---|---|---|---|---|
| | | | (Null Model) | AOR (95%CI) | AOR (95%CI) | AOR (95%CI) |
| **Women education level** | Did not attend education | 1 | | 1 | | 1 |
| | Complete primary education | 1.70(0.88,3.28) | | 1.45 (0.69,3.02) | | 1.54 (0.74,3.21) |
| | Complete secondary education and above | 3.61(1.78,7.32) | | 2.93(1.18,7.28) | | 3.15(1.25,7.89) |
| **Husband education level** | Did not attend education | 1 | | 1 | | 1 |
| | Complete primary education | 1.43(0.74,2.78) | | 0.93(0.43,1.96) | | 0.87(0.41,1.83) |
| | Complete secondary education and above | 3.48(1.68,7.21) | | 1.63(0.63,4.18) | | 1.39(0.53,3.62) |
| **Being a member in WDA** | Yes | 3.84(2.16, 6.83) | | 2.85(1.53,5.33) | | 2.91(1.56,5.44) |
| | No | 1 | | 1 | | 1 |
| **HEWs home visiting** | Yes | 1.85(1.04,3.28) | | 1.91(1.03,3.57) | | 1.69(0.88,3.24) |
| | No | 1 | | 1 | | |
| **Being insured** | Yes | 3.07(1.65,5.71) | | 2.52(1.29,488) | | 2.59(1.33,5.01) |
| | No | 1 | | 1 | | |
| **Getting health education** | Yes | 1.87(1.06,3.29) | | 2.40(1.314.42) | | 2.44(1.33,4.45) |
| | No | 1 | | 1 | | |
| **Facility readiness** | Yes | 1.66(0.52,5.23) | | | 0.78(0.27,2.26) | 0.54(0.19,1.51) |
| | No | 1 | | | 1 | 1 |
| **Distance from home to health facility** | <=5km | 4.37 (1.23,15.29) | | | 5.09 (1.52,16.99) | 4.81 (1.55,14.95) |
| | >5km | 1 | | | 1 | 1 |
| **Mass-media exposure** | High | 1.89(0.72,4.98) | | | 2.20(0.96,5.03) | 2.39(1.11,5.15) |
| | Low | 1 | | | 1 | 1 |
| **Random effects** | Variance | | 0.43 | 0.37 | 0.1 | 0.03 |
| | ICC (%) | | 12 | 10 | 3 | 0.9 |
| | PCV (%) | | ref | 14 | 76 | 93 |
| | AIC | | 401.43 | 365.38 | 398.52 | 362.62 |
| | -2Loglikelihood | | 397.44 | 345.38 | 388.52 | 336.62 |

of multivariable multilevel logistic regression analysis indicates that at the individual level, completing secondary and above education, being a member of a WDA, getting health education from a health post, and being a member of community-based health insurance; and at the community level, distance from home to health facility, and mass-media exposure were significant predictors of women's completion of the maternal continuum of care.

According to our review of literature, the proportion of women completing the maternal continuum of care ranges from 6.4% [26] to 67.8% [27]. The current study was congruent with research studies conducted in Ghana [28, 29]. However, our finding was lower as compared to findings of studies conducted in other parts of Ethiopia [22, 27, 30–34], and elsewhere in Tanzania [35], and nine South Asian and sub-Saharan African countries [36].

The deflated proportion of women in the current study as compared to other studies could be attributed to the fact that our study considered women who had received at least 4 ANC visits, SBA at birth, and PNC visits within 48 hours after birth, whereas others included the postpartum period within six weeks after birth [28–30, 32, 34, 37]. The larger drop out in women's completion of the maternal continuum of care puts women at higher risk of unwanted death, as women could miss continuity in maternal healthcare utilization. Another justification may

be the low rate of early initiation and completion of ANC visits [38–40]. It is believed that early booking of ANC creates an opportunity for a woman to get pregnancy-related information such as birth preparedness, danger signs during pregnancy and the postpartum period, nutritional counseling, and complete content of focused ANC components and helps with early checkups for medical and obstetric conditions [32, 40–42]. Another possible justification could be that in the current study, the rate of late initiation of ANC (after 16 weeks of pregnancy) was nearly 50%, i.e., one in every two women did not receive an early ANC visit, as women have not been provided the aforementioned healthcare benefits of early initiation of ANC visits. Therefore, we suggest early booking of ANC and staying on the path of the maternal continuum of care to improve women's and newborns' well-being. The low point of women's completion of the maternal continuum of care may be explained by the fact that though "home delivery free" is the program that has been launched by the government of Ethiopia, it is still one of the unresolved challenges of maternal health program indicators. For example, the Ethiopian demographic survey of 2016 [9] showed that 73% of births occur at home. In our study, among women who did not receive facility delivery and PNC visits, nearly two-thirds of women preferred Traditional Birth Attendants (TBA) to give birth and did not have awareness about the importance of PNC after birth. They perceived that they and their babies were safe and did not intend to go to a health facility. Another reason might be that health-seeking behavior, beliefs, and norms about the place of birth could contribute to a low rate of women's completion of the maternal continuum of care. Literature states that the most common reasons for home delivery are family and relatives influence, usual practice, unexpected labor, and not being sick, depending on traditional healers and spiritual healing, such as prayer [43]. Women's perception that pregnancy is not an illness and that attending ANC is useless could act as a barrier to utilizing healthcare services [44]. Moreover, unlike other studies [27, 30–32, 34, 36, 37, 45, 46] that reported higher proportions of the maternal continuum of care by using ordinary logistic regression, our study considered multilevel multivariable logistic regression models.

Regarding predictors, our finding revealed that the odds of women's completion of secondary and above education had a significant association with women's completion of the maternal continuum of care. This finding aligns with the findings of studies conducted in Ethiopia [22, 30, 31, 47] and abroad [29, 36, 37, 48, 49]. The possible justification may be that educated women could easily grasp pregnancy-related counseling during healthcare provision and information through mass media. Moreover, educated women may have a role in self-determination so that they can decide by themselves to seek healthcare from healthcare providers. In addition, educated women may not be influenced by beliefs and norms. A study conducted in rural Mali on the role of beliefs and norms for maternal healthcare utilization revealed that women were expected to obey their husbands during childbirth in order to have an easy delivery [50]. If the husband orders the woman to stay at home during labor, the woman should obey him and stay at home for home delivery.

In this study, women who were members of a WDA were more likely to complete the maternal continuum of care as compared to their counterparts. This finding was supported by studies done in Ethiopia [51, 52]. The similarity of our findings might be that an active woman in a WDA could get different benefits in the kebele, such as health education about birth preparedness and complication readiness, being linked to health facilities by HEWs at the time of referral, reducing delays that could be related to a pregnant woman, and enhancing harmonized relationships with HEWs, community, and healthcare facilities.

The odds of women completing the maternal continuum of care were higher among respondents who were members of community-based health insurance as compared to those who were not insured. Our finding was in concord with other studies conducted in Ethiopia

[53, 54], Ghana [55], Manzi [56], Nigeria [57], and Tanzania [58]. Ethiopia has been implementing the Community Based Health Insurance (CBHI) scheme since 2011 with a vision of promoting health of poor rural residents [59, 60], as the CBHI package benefits a woman to get all family health services which is part of Ethiopia's essential health package [61]. The association of CBHI and maternal continuum of care in the current study might be due to the fact that the existing CBHI scheme in Ethiopia provides maternal healthcare services for free to all women [62]. Due to the low economic conditions of the country, women could be less motivated to use maternal healthcare services if they were required to pay for them. Another possible explanation might be that community health insurance reform in Ethiopia could have had an influence on the health-seeking behavior of a woman [53]. Moreover, CBHI enrolment rate has been growing in Ethiopia, particularly the highest rate was recorded in Amhara region where the study was conducted [60].

The findings of this study showed that getting health education at the health post level made women more likely to complete the maternal continuum of care as compared to women who did not get health education. Our finding was consistent with a prior study [63] and a study done in Pakistan revealed that facility delivery was higher among women who got health education from community health workers when compared to those who did not get health education during the pregnancy period [64]. In Ethiopia, frontline HEWs are commonly tasked with providing maternal and child health services at primary healthcare units [65]. HEWs' health message advocacy for maternity service utilization at the health post level could have an influence on the maternal healthcare continuum. This finding was reaffirmed by [66], as community health workers have a vital role in facilitating women's completion of maternal continuum of care.

Regarding community level distance, the odds of completing the maternal continuum of care were higher in women who had a walking distance of less than or equal to five kilometers from home to reach health facility as compared to their counterparts. Studies conducted in other parts of Ethiopia [30, 31, 67] and elsewhere [28, 63, 68–70] supported this finding that less travel time to health facilities predicted completion of the maternal continuum of care. This could be because women who live in difficult-to-reach areas or are unable to access health facilities may face lack of transportation to health facilities, particularly at night [71] as many times labor begins at night [72] and most women take walks by foot or animal; lack of transportation and long distances are barriers for maternal healthcare utilization [73]. Another reason might be that women at remote locations could have less awareness and miss a chance to get pregnancy-related information as compared to women who live near a country/health facility. Another possible explanation might be the inaccessibility of health facilities, according to Berhan et al., as women pay around 4000 Birr for less than a hundred kilometers of travel in Ethiopia, which is unfair and too high in the country by any standard [74].

This study found that women who were exposed to the mass media were more likely to complete the maternal continuum of care than those women who were not exposed. This finding is in line with previous studies done in Ethiopia [29, 37]. According to a study in Pakistan, about 58% of women with weekly exposure to mass media gave birth in health institutions as compared to 36% of women with less frequent mass-media exposure [64]. The possible explanation for this is that the mass media has the potential to influence developing positive behavior towards maternal health service utilization because women can easily access maternal and child health-related information at home.

## Strengths and limitations of the study

First, we carried out a community-based study to understand the current level of maternal continuum of care and its associated factors using individual and community level variables.

Second, the sample size we used in this study was yielded after a thorough estimation of sample sizes using proportions of focused ANC visits, SBA, and continuum of maternal care. Third, the use of a simple random sampling technique in this study has contributed to the strength of the study. Fourth, we provided adequate training and close supervision during field activities. Fifth, each estimate was tested using the demand and supply side inquiries, which can be reflected as an actual problem of women's completion of maternal continuum of care.

Recall bias may be the limitation of this study, as the participants might not remember the previous event other than recent events. Nevertheless, we attempted to specify questions related to the service given during prenatal, natal, and postnatal periods by probing. In our study, we did not exclude women who had previous pregnancy complications; this could have introduced social desirability bias in which respondents may have answered questions that they thought would lead to their being accepted. However, during the response, an open-ended approach was used to prevent the participants from simply agreeing or disagreeing. Owing to resource constraints, the study was limited in a small locality and this might make it to fall short of generalizability to a wider population. Follow-up studies covering larger segments of population and focusing on other contextual factors beyond the individual and community level variables could give better findings for the improvement of maternal continuum of care. Moreover, we tried to control confounding through maximizing potential variables to be included in the study and applying multivariable logistic regression during analysis stage.

## Conclusions

Women's completion of the maternal continuum of care was extremely low in the study area. There was a substantial reduction in community variation in the final model, indicating that there was a reduced representation of unobserved variables that explain the variation. The findings suggest that women's education, being insured, being a member of a WDA, health education at the health post level, distance from home to health facility, and women's exposure to mass media are contextual factors contributing to women's completion of the maternal continuum of care. Therefore, the existing health system should consider multilevel intervention strategies that focus on providing maternal health education, facilitating insurance mechanisms, encouraging women's participation in health clubs, and ensuring physical accessibility to healthcare facilities to be more effective in improving maternal health services.

## Supporting information

**S1 Dataset. Dataset of why maternal continuum of care remains low in Northwest Ethiopia? A multilevel analysis.**
(DTA)

## Acknowledgments

We thank the Amhara national regional state health bureau, Central Gondar zone health department, Wogera and Gondar Zuriya district health offices, the University of Gondar Comprehensive Specialized Hospital, and HEWs for their provision of necessary information and support during data collection. Our gratitude also goes to the study participants, data collectors, and supervisors who took part in the study.

## Author Contributions

**Conceptualization:** Tesfahun Hailemariam, Asmamaw Atnafu, Lemma Derseh Gezie, Binyam Tilahun.

**Data curation:** Tesfahun Hailemariam, Asmamaw Atnafu, Lemma Derseh Gezie, Binyam Tilahun.

**Formal analysis:** Tesfahun Hailemariam, Asmamaw Atnafu, Lemma Derseh Gezie, Binyam Tilahun.

**Investigation:** Tesfahun Hailemariam, Asmamaw Atnafu, Lemma Derseh Gezie, Binyam Tilahun.

**Methodology:** Tesfahun Hailemariam, Asmamaw Atnafu, Lemma Derseh Gezie, Binyam Tilahun.

**Project administration:** Tesfahun Hailemariam, Asmamaw Atnafu, Lemma Derseh Gezie, Binyam Tilahun.

**Resources:** Tesfahun Hailemariam, Asmamaw Atnafu, Lemma Derseh Gezie, Binyam Tilahun.

**Software:** Tesfahun Hailemariam.

**Supervision:** Tesfahun Hailemariam, Asmamaw Atnafu, Lemma Derseh Gezie, Binyam Tilahun.

**Validation:** Tesfahun Hailemariam.

**Writing – original draft:** Tesfahun Hailemariam, Asmamaw Atnafu, Lemma Derseh Gezie, Binyam Tilahun.

**Writing – review & editing:** Tesfahun Hailemariam, Asmamaw Atnafu, Lemma Derseh Gezie, Binyam Tilahun.

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
