## [Decision Letter · Decision Letter 0]

12 Jul 2022

PONE-D-21-38409Why maternal continuum of care remains low in Northwest Ethiopia? a multilevel logistic regression analysisPLOS ONE

Dear Dr.Tesfahun Hailemariam,

Thank you for submitting your manuscript to PLOS ONE. After careful consideration, we feel that it has merit but does not fully meet PLOS ONE’s publication criteria as it currently stands. Therefore, we invite you to submit a revised version of the manuscript that addresses the points raised during the review process.

We look forward to receiving your revised manuscript.

Kind regards,

Sebsibe Tadesse, PhD

Academic Editor

PLOS ONE

Journal Requirements:

2. You indicated that you had ethical approval for your study. In your Methods section, please ensure you have also stated whether you obtained consent from parents or guardians of the minors included in the study or whether the research ethics committee or IRB specifically waived the need for their consent

Reviewers' comments:

Reviewer's Responses to Questions

**Comments to the Author**

1. Is the manuscript technically sound, and do the data support the conclusions?

Reviewer #1: Yes

Reviewer #2: Yes

2. Has the statistical analysis been performed appropriately and rigorously? 

Reviewer #1: Yes

Reviewer #2: Yes

3. Have the authors made all data underlying the findings in their manuscript fully available?

Reviewer #1: Yes

Reviewer #2: Yes

4. Is the manuscript presented in an intelligible fashion and written in standard English?

Reviewer #1: Yes

Reviewer #2: Yes

5. Review Comments to the Author

Reviewer #1: Overall impression

The paper is clearly and logically written, well organized, and easy to follow. The introduction provides the reader useful background and context that sets up the rest of the paper well. The authors present robust and novel findings that address the factors that affect women’s completion of the maternal continuum of care in the Wogera and Gondar districts through a community-based cross-sectional study. The methods were clearly defined, and the results and discussions well outlined and supported with existing evidence and past studies in Ethiopia. The paper also did well to adjust for the design effect by assessing various levels of intraclass correlation coefficients using four different models.

The paper requires major and minor revisions including the revision of wording choice and consistency, the justification of the interview tool and sampling technique, addition of citations, and the expansion on the discussion of study limitations. Moreover, I would advise the author to complete a grammar revision of the paper to improve the flow and readability of the text and revise the structure of some of the sections within the paper.

Detailed Revision:

Please refer to the attached 'Reviewer's Detailed Comments' document for further detailed recommendations.

Reviewer #2: The findings of this study has significant input to answer some questions regarding to poor maternal health care in Ethiopia. The researcher also tried to mention some findings which are highly related to this topic and he tried to mention the gap which has been not addressed.

Data analysis procedure as well as assumptions which are considered during data analysis were clearly mentioned.

6. PLOS authors have the option to publish the peer review history of their article (what does this mean?). If published, this will include your full peer review and any attached files.

Reviewer #1: **Yes: **Lena Kan

Reviewer #2: No

---

## [Author Response · Author response to Decision Letter 0]

14 Aug 2022

Date 20/07/2022

Rebuttal Letter

Subject: Question-by-question responses

Manuscript title: Why maternal continuum of care remains low in Northwest Ethiopia? 

a multilevel logistic regression analysis

Manuscript number: PONE-D-21-38409

Authors: Tesfahun Hailemariam, Asmamaw Atnafu, Lemma Derseh Gezie, and Binyam Tilahun

Dear reviewers 

Thank you for taking the time to review and comment upon our manuscript, (Manuscript number: PONE-D-21-38409, Why maternal continuum of care remains low in Northwest Ethiopia? a multilevel logistic regression analysis). We found the advice constructive and have incorporated the suggestions into our revised or cleaned manuscript. We have responded to each of the reviewers’ comments in our comment-by-comment responses below. It is our belief that the manuscript is substantially improved after all those comments and suggestions were addressed. We noted that the corrected texts at cleaned version may not exactly found on the page and line as mentioned by reviewers at the original commented manuscript because of the extensive revisions of each page and line. We want to inform you that the corrections made in the main document are indicated with blue color. 

Thank you again for your thoughtful comments.

On behalf of the group of authors

Sincerely,

Tesfahun Hailemariam

Reviewers' comments to author:

The paper is clearly and logically written, well organized, and easy to follow. The introduction provides the reader useful background and context that sets up the rest of the paper well. The authors present robust and novel findings that address the factors that affect women’s completion of the maternal continuum of care in the Wogera and Gondar districts through a community-based cross-sectional study. The methods were clearly defined, and the results and discussions well outlined and supported with existing evidence and past studies in Ethiopia. The paper also did well to adjust for the design effect by assessing various levels of intraclass correlation coefficients using four different models. 

The paper requires major and minor revisions including the revision of wording choice and consistency, the justification of the interview tool and sampling technique, addition of citations, and the expansion on the discussion of study limitations. Moreover, I would advise the author to complete a grammar revision of the paper to improve the flow and readability of the text and revise the structure of some of the sections within the paper. 

Comment: Revision of wording choice and consistency

Response: Thank you. We have revised the paper for its word choice and consistency according to the comment given.

Comment: The justification of the interview tool and sampling technique

Response: Thank you so much. The interview tool was developed in English, after thorough literature review, and translated into the local language (Amharic). Since expert’s view was sought for psychometric properties of face and content validity, experts (Gynecology and obstetrics specialist (1), research and publication directorate (1), Health Informatics specialist (2), Midwifery specialist (2), Emergency and obstetric officer specialist (1) were invited to review the relevance of each question in the interview tool. The tool was revised according to expertise’s view and piloted before data collection was started. The proportion of rated item and scale content validity index was computed and experts rated item content validity index and scale level validity index were 0.98 and 0.81, respectively. For wealth index assessment, principal component analysis was employed to reduce data that measure the same construct together and the data was recoded into binary variables. Variables with low frequencies were combined together and frequency more than 95% and less than 5% were excluded from the analysis. Under the components loading, there were twenty-one variables correlated with measuring the wealth index such as having own house, toilet, electricity, kitchen, charcoal, television, radio, modern bed, cotton, mobile phone, farming land, having animals in the house such as plough, house, cows, horse donkey, goat, sheep, chicken, having separate room for animals, having beehive, having bank account number with deposited Birr (Ethiopian currency). Scale reliability coefficient and Kaiser-Meyer-Olkin measure of sampling adequacy of variables were 0.72 and the 0.80, respectively. Wealth quantile was categorized into five categories such as poorest, poor, middle, richer and richest. Data collectors and supervisors were trained on the instrument. The household data were collected, using an interviewer administrated questionnaire via face-to-face interviews.

Regarding to sampling technique, in this study, we used a multi-stage sampling technique to reach study participants. Wogera and Gondar zuriya districts (the districts where our study conducted) were selected randomly among the six transformation woredas in the central Gondar zone, Northwest Ethiopia. Specifically, the primary sampling units were kebeles with the respective health centers while the secondary sampling units were women who gave birth during the past 12 months. The health centers (or the respective kebeles) were selected with a simple random sampling technique. And to select sample participants, a sampling frame of mothers was developed using a list of eligible women from the community health information system family folder(pouch). Then the sampling interval was obtained by dividing the source population (8393) by the estimated total sample size (811). The first study participant (the 2nd mother) was selected by a simple random sampling technique from the first sampling interval, and all other mothers were selected systematically by taking every 10thmother in the frame. When two or more women were found at a single household level, one of them was selected randomly and included in the study. Seriously ill or women who were unable to speak were excluded from the study. For those women who were selected for an interview and were not available during data collection, we waited for them for three to five days to get them. If they were not still accessible, they would be considered non-respondent. 

Comment: Addition of citations, and the expansion on the discussion of study limitations. 

Response: Thank you so much. We added recent references and expanded study limitations in the discussion section. 

Question: Moreover, I would advise the author to complete a grammar revision of the paper to improve the flow and readability of the text and revise the structure of some of the sections within the paper. 

Response: Thank you so much. Critical refinements were made to improve further its English and revisions on the structure of the paper.

Please refer to the minor revisions for detailed recommendations. 

Minor revisions: 

1. The paper needs more discussion of study limitations which may include expanding on: 

Comment: Selection bias: error due to random chance, bias from selecting the 11 health centers (clusters) and kebeles for the study; limited external validity: study only conducted in 2 districts in Ethiopia 

Response: Thank you so much. In order to minimize the selection bias that is possible from selecting the 11 health centers (clusters) and kebeles for the study, we have applied probability sampling technique using the defined target population and sampling frame. Owing to resource restrictions, the study was limited in a small locality (Wogera and Gondar Zuriya districts) and this might make it to fall short of generalizability to a wider population. Longitudinal studies covering larger segments of populations and focusing on other contextual factors beyond the individual and community level variables would give better findings for the improvement of maternal continuum of care. We have included this limitation in the discussion section of limitation part. We feel that with its limitation the study could be interpreted/compared to other similar settings. 

Question: Confounding bias: any other unknown or unmeasured confounding factors that were not controlled for 

 Response: Thank you so much. We have exhaustively reviewed literatures and included the potential variables in the study in order to minimize biases that could be arised due to confounding. The random effect we applied in this study could capture the effect of unaccounted variables beyond the individual variables and may quantify the bias estimate related with third variable. Moreover, the conventional alpha 0.05 significance wouldn’t report the errors in the estimates so that the estimate could be over or under estimated. The findings in our study were presented with adjusted odds ratio which reduces systematic and random error misclassification bias.

Minor revisions: 

1. Paper Section: Overall

Suggestion: Keeping the word choice and labeling consistent 

Specific example

Utilize relevant and defined acronyms consistently throughout paper: 

Comment: Page 3, Line 17: antenatal care was defined as ANC. Suggest using “ANC” consistently throughout rest of the paper 

Response: Thank you so much. We have used “ANC” consistently for antenatal care throughout the paper. 

Comment: Suggest assigning acronym “TBA” to ‘Traditional birth attendants” earlier in the paper and standardizing use of “TBA” throughout the rest of paper 

Response: Thank you so much. We have used “TBA” early in the paper for Traditional Birth Attendant and used the acronym throughout the paper according to the comment. 

Comment: Suggest assigning acronym “SBA” to “Skilled birth attendants” early in the paper and standardizing use of “SBA” throughout rest of paper

Response: Thank you so much. We have used “SBA” for Skilled Birth Attendant early in the paper and we have applied it throughout the paper. 

Comment: Suggest assigning acronym “WDA” to “Women’s Development Army” earlier in the paper, as Table 3 references “WDA” without the acronym being defined earlier and standardizing use of WDA in the rest of paper

Response: Thank you so much. We have abbreviated “WDA” early in the paper and used the acronym throughout the paper as per the comment. 

Comment: Suggest assigning acronym “HEW” to “Health Extension Worker” earlier in the paper (before page 20) since Table 3 references ‘HEW” without the acronym being defined earlier and standardizing use of HEW throughout the rest of the paper 

Response: Thank you so much again. We have abbreviated “HEWs” early in the paper and used the acronym throughout the paper. 

Comment: Suggest assigning acronym “PNC” to postnatal care and standardizing use of PNC throughout the rest of the paper

Response: Thank you so much. We have used “PNC” early in the paper and used it throughout the paper as per the comment given. 

Comment: Intraclass correlation was assigned acronym “ICC” on Page 9, Line 12. No need to define the acronym again on Page 9, Line 21. Suggest using “ICC” consistently throughout rest of the paper 

Response: Thank you so much. We have made correction on the manuscript based on the comment, using ICC consistently throughout the paper.

Comment: Suggest revising “antenatal care four and above” to “at least 4 ANC visits” 

o Page 5, Lines 23-24

o Page 8, Lines 12-13 

o Page 13, Line 5 

o Page 14, Line 1

o Page 18, Line 10 

Response: Thank you so much. We have revised “antenatal care four and above” to “at least 4 ANC visits” consistently throughout the paper. 

2. Abstract 

 Word choice

Comment: Page 2, Line 3: suggest replacing “intractable” to “significant”

Response: Thank you so much. We have replaced an “intractable” word to “significant”

3. Introduction 

Wording clarification and consistency 

Suggest revising the following sentences

Comment: Page 3, Line 4-6: clearly explain what the numbers represent (e.g., number of deaths, life-time risk). Also, please clarify which year these statistics were estimated in. Page 3, Line 10: suggest adding year of estimation.

Response: Thank you so much. We have explained the numbers presented at page 3 line 4-6 and the year in which the statistics were estimated.

Comment: Page 5, Line 3-4: please clarify what you mean by “women with less than one year” – what is the less than one year referring to? 

Response: Thank you so much. We have made a clarification of the word “women with less than one year.” The author has made change on the paper as “women with a child less than one year”. 

Wording choice and consistency: 

Comment: Page 4, Line 1: suggest changing “skilled healthcare attendant” to “skilled birth attendant” (SBA) to align with wording in rest of paper 

Response: Thank you so much. We have changed skilled healthcare attendant to “skilled birth attendant” (SBA) in page 4 line 1 and aligned it in all parts of the paper where SBA is mentioned. 

Comment: Page 4, Line 11: suggest changing “besides” to “moreover” 

Response: Thank you so much. We have changed the word “besides” to “moreover” as per you suggestion.

Comment: Page 4, Line 16: suggest revising “studies” to “past/recent studies” 

Response: Thank you so much. We have revised the word “studies” to “past studies” according to the comment given.

4. Methods 

Sample size clarification 

Consider including a sample size calculation table in Appendix: 

Comment: Suggest presenting sample size calculation matrix table under different scenarios (and assumptions) and justification of the chosen sample size

Response: Thank you so much. Our study was based on the following sample sizes estimated using different assumptions as presented in the following calculation matrix table. 

Table 1: Sample size calculation for the study objective using single population proportion formula and considering factor affecting the outcome variable 

No 

Variables Study

In 95% CL 80% power OR Exposed (%) Unexposed (%) Design effect Ratio Sample size NR- rate Total 

Sample size

 Proportion of focused antenatal care (39.9%) (22) 95% - - - - 2 - 736.9 10% 811

 Proportion of SBA after completing at least 4 ANC visits (31.1%) (22) 95% - - - - 2 - 658.5 10% 724

 Proportion of women retained in the continuum of maternal care (12.1%) (22) 95% - - - - 2 - 326.8 10% 360

 Associated factors with CoC Being a model family in the community (23) 95% 80% 2.13 49.7 50.3 2 1:1 254 10% 559

NB: Finally, after internal comparison, we have considered the largest sample size (811) and used it to community based cross-sectional study in order to assess women’s completion of maternity continuum of care. 

Clarify age range 

Comment: Page 5, line 14: Please clarify what the age range for reproductive age group is (e.g., 15-54) and explain why this range was chosen. 

Response: Thank you so much. For this study, women of reproductive age group (15-49) were selected. Though the age range for reproduction is possible out of the mentioned age range, the reproductive age group in Ethiopia (15-49) was sought our attention in order to assess the prevalence and its associated factors of maternal continuum of care using both individual and community level variables. The current Ethiopian demographic health survey [1] also focuses the age range of women 15-49 as the majority of women fall in this age ranges. Other study also supports these evidences [2]. 

Clarify outcome variable 

Comment: Page 9, Line 3: Suggest clarifying that the outcome variable is the “maternal continuum of care” (binary). 

Response: Thank you so much. Maternal continuum of care contains three combined maternity care levels, first: women with antenatal care one to antenatal care four and above; Second: women who were completed antenatal care 4 and above to skilled birth attend, and third: women from skilled birth attendant to postnatal care visit within 48 hours after delivery. Hence, our study considered maternal continuum of care “1” if a woman received the three combined service together, and “0” otherwise. 

5. Results

Revise Tables

Comment: Tables 1-2: Suggest removing column “Total (%)”: it is not contributing more useful information

Response: Thank you so much. We have removed table 1 and table 2 total (%) column as per the suggestion given in page 10 and 11, and 12 and 13 respectively. 

Comment: Tables 1-3: Make sure that spacing is consistent in table 1 (e.g., N (%)) 

Response: Thank you. We have made consistent spacing in both “Yes” and “No” column in page 10 and 11.

Wording choice 

Comment: Page 11, suggest revising “intendedness of pregnancy” to “pregnancy intention” 

Response: Thank you so much. We have replaced “pregnancy intendedness” to “pregnancy intention”, page 11.

Wording choice and consistency: 

Comment: Standardize capitalization of “Model I-IV” in Results section,

o Page 14, Line 22: Capitalize “model II” 

o Page 15, Line 1: Capitalize “model III”

o Page 15, Line 4: Capitalize “model IV” 

Response: Thank you so much. We have capitalized Model I-IV in result section in page 14 and 15, and throughout the paper as well.

Comment: Clarify “counterparts” when summarizing Odds Ratios and comparisons 

Suggest revising “counterparts” to “lower-education counterparts” (Page 15, Line 11) 

Response: Thank you so much. We have revised the word “counterparts” with “lower-education counterparts”.

Missing results 

Comment: Page 14, Line 17: fill in the blank: what = 0.43? 

Response: Thank you so much. We have filled the missing result, page 14, line 17

6. Discussion 

Add citations 

Comment: Page 17, Line 3-4: add citations for statistics “one of every fourteen” and “one out of eight”

Response: Thank you so much. In the first paragraph of the discussion section, we outlined our summary findings. One of the findings is maternal continuum of care as it was presented in Fig 1, i.e., 6.9% of women compete maternal continuum of care. This was explained in the discussion section in the form “one from every fourteen women”. The other finding indicates that from all the women who were surveyed at the study districts, about 12.5% of women didn’t get any antenatal care visit during pregnancy life time from existing health system. This was also explained in discussion section in the form of “one out of eight women” were not getting maternity service from existing health system. These findings were explained in detail in result section at “dropouts of women completion of maternal continuum of care” part, thus, we feel that it does not need citations as they are the findings of our/current study. 

Comment: Page 19, Lines 16, 19-20: add citations for the community-based health insurance scheme/reform in Ethiopia

Response: Thank you so much. We added citations for the community-based health insurance scheme/reform in Ethiopia.

Wording choice and consistency and clarification 

Wording choice and consistency: 

Comment: Suggest revising Page 18, Lines 12-13 to “puts women at higher risk of unwanted death”

Response: Thank you so much. As per your suggestion, we have revised the word “risks women to unwanted death” to “puts women at higher risk of unwanted death”

Comment: Suggest revising Page 19, Lines 18-19 to “aligns” 

Response: Thank you so much. We have revised the word “is in consonance” to “aligns” according to your comment. 

Wording clarification: 

Comment: Please clarify what “elsewhere” means (Page 19, Line 19) 

Response: Thank you so much. In our study we used the word elsewhere to indicate countries where the primary studies were conducted (Ghana (Ref 30), South Asia and sub-Saharan Africa (Ref 38), Nepal (Ref 40), Cambodia(Ref 42), and Egypt (Ref 44). 

Clarification needed 

Comment: Please justify why Pakistan and other countries (e.g., Cambodia, Nepal, Lao PDR, Egypt) were included as comparisons of interest to the results of the study. Would suggest removing these comparisons and only keeping references in Ethiopia. 

Response: Thank you so much. In this section we have attempted to compare utilization of maternal continuum of care at different countries which have different context. However, the variation of our finding from previous finding conducted in Africa and other countries were justified why is low or high using the existing evidences. The only thing comparing our finding in this section with studies conducted in some other countries like Cambodia, Egypt, Lao etc is that to show the existing evidence or what is known regarding to our findings. According to the comment given, we removed references and tried to keep the references in Ethiopia. 

Strengths & limitation 

Comment: Strengths: No strengths were outlined in the Discussion section 

Response: Thank you so much. We have included the strengths section in the discussion part of the paper. 

Comment: Limitations: to consider further limitations in the study 

Response: 

Comment: Consider selection bias: error due to random chance, bias from selecting the 11 health centers (clusters) and kebeles for the study;

Response: Thank you so much. In order to minimize the selection bias that is possible from selecting the 11 health centers (clusters) and kebeles for the study, we have applied probability sampling technique using the defined target population and sampling frame. 

Comment: limited external validity: study only conducted in 2 districts in Ethiopia. Explain the relevance of the study beyond the context of Ethiopia. 

Response: Thank you so much. Owing to resource restrictions, the study was limited in a small locality (Wogera and Gondar Zuriya districts) and this might make it to fall short of generalizability to a wider population. Longitudinal studies covering larger segments of populations and focusing on other contextual factors beyond the individual and community level variables would give better findings for the improvement of maternal continuum of care. We have included this limitation in the discussion section, at limitation part. We feel that with its limitation the study could be interpreted/compared to other similar settings. 

Comment: Consider confounding bias: any other unknown or unmeasured confounding factors that were not controlled for 

Response: Thank you so much. We have exhaustively reviewed literatures and used the potential variables that to be included in the study in order to minimize biases that could arise due to confounding. The random effect we applied in order to capture the effect of unaccounted variables, beyond the individual variables may quantify the bias estimate related with third variable. Moreover, it is obvious that the conventional alpha 0.05 significance wouldn’t report the errors in the estimates so that the estimate could be over or under estimated, therefore, the findings in our study were presented with adjusted odds ratio with systematic and random error misclassification bias. 

Comment: Study instrument: clarify whether tool is standardized in Ethiopia / beyond; explain who the “expert reviewers” were 

Response: Thank you so much. This section was elaborated in detail in the first page in similar comment section. The explanation of who the expert reviewers were elaborated at the instrument section of the paper. We applied both face and content validity of the tool in which expert judgment on what the instrument is looks like: wording, layout, clarity, comprehensiveness of the tool was checked in face validity and content validiy in which using content validity index. The mean proportion of experts rated item relevant is in a range (I-CVI)=0.98 and S-CVI/UA= 0.81, 98% of the total items are judged content valid. Accordingly, the tool was checked for its appropriateness and revision was also undertaken.

7. Grammar 

Grammar 

Comment: Please revise comments throughout manuscript (attached PDF). 

Response: Thank you so much. The whole manuscript sections were revised and refinement was made. The cleaned version and manuscript with comment addressed in blue color were attached in the submission system.

Thank you again for the effort you made to review our manuscript!

1. Mini E, Demographic E. health survey 2019: key indicators report. The DHS Program ICF. 2019.

2. Yeshaw Y, Kebede SA, Liyew AM, et al. Determinants of overweight/obesity among

reproductive age group women in Ethiopia: multilevel analysis

of Ethiopian demographic and health survey. BMJ Open 2020;10:e034963. doi:10.1136/ bmjopen-2019-034963

---

## [Editor Report · Decision Letter 1]

4 Sep 2022

Why maternal continuum of care remains low in Northwest Ethiopia? a multilevel logistic regression analysis

PONE-D-21-38409R1

Dear Dr. Tesfahun Hailemariam,

We’re pleased to inform you that your manuscript has been judged scientifically suitable for publication and will be formally accepted for publication once it meets all outstanding technical requirements.

Kind regards,

Sebsibe Tadesse, PhD

Academic Editor

PLOS ONE

---

## [Editor Report · Acceptance letter]

8 Sep 2022

PONE-D-21-38409R1 

Why maternal continuum of care remains low in Northwest Ethiopia? a multilevel logistic regression analysis

Dear Dr. Hailemariam:

I'm pleased to inform you that your manuscript has been deemed suitable for publication in PLOS ONE. Congratulations! Your manuscript is now with our production department. 

Kind regards, 

on behalf of

Dr. Sebsibe Tadesse 

Academic Editor

PLOS ONE